# "I Thought I Was Going to Die like Him": Racial Authoritarianism and the Afterlife of George Floyd in the United States and Brazil

**Jaimee A. Swift**

Department of Political Science, James Madison University, Harrisonburg, VA 22807, USA; swiftja@jmu.edu

**Abstract:** This paper offers a brief yet comprehensive comparative analysis of historical and contemporary racial authoritarian violence in the United States and Brazil. Utilizing Black feminist historian and literary scholar Saidiya Hartman's theorization of the "afterlife of slavery" and Michael Dawson's linked fate, I examine how the processes of racialization and the racial logics of subordination have and continue to shape the contours of Black life in the United States and in Brazil. Moreover, in this work, I interrogate the afterlife of George Floyd and the afterlives of Black Brazilian victims and survivors of racial authoritarian violence; the political, transnational, and symbolic impacts of Floyd's death; and Diasporic understandings of linked fate on racial authoritarian violence between Black communities in the United States and in Brazil.

**Keywords:** racial authoritarianism; anti-black police violence; United States; Brazil; George Floyd

## 1. Introduction: *"I Thought I Was Going to Die like Him":* The Afterlives of George Floyd and Linked Fate in Brazil

On 25 May 2020, George Floyd, a 46-year-old Black American man, was arrested by four Minneapolis, Minnesota police officers after a convenience store clerk alleged that he had purchased cigarettes with a counterfeit USD 20 bill (Hill et al. 2020). Handcuffed while lying prone on the ground, a video recording of Floyd's arrest showed him pleading for help and screaming, "I can't breathe", as Derek Chauvin, a White police officer, pinned his knee on Floyd's neck for over nine minutes. After desperately saying over 20 times that he could not breathe, Floyd later died from asphyxia.[1] His death, which occurred during the third year of right-wing, conservative, U.S. President Donald Trump's term, catalyzed countless protests to end anti-Black police violence and brutality in the United States and globally. On 30 May 2020, five days after Floyd's death, a similar incident happened to a 51-year-old Black woman in Parelheiros, one of the 96 districts in the south of São Paulo, Brazil. The mobile video footage, which was released on 12 July 2020, on Fantástico, a news program on the Brazilian network, TV Globo, showed a "[military] police officer pinning a woman to the ground by pressing his foot on her neck. He then lifts up his other foot, placing all his weight on the victim's neck as she lies face down" (France24 2020).

Although the victim was not publicly identified, she was a middle-aged, small bar owner operating her business during the COVID-19 pandemic, when the Brazilian government told businesses to cease operations (BBC 2020). Prior to being stomped by the military police officer, she was punched several times, to the point of falling and breaking her leg. While the bar owner's friend was being pinned down by a police officer, cellphone footage filmed by onlookers showed another officer "… pointing a gun at another man, who [was] standing in the middle of the street. The man peels off his T-shirt and raises his arms in the air, as if to surrender. The police officer then joins his fellow officer on the

sidewalk, pushing the owner of the bar away from the man pinned to the ground" (France24 2020). In her pursuit to defend her friend who was being harassed by police, the Black woman bar owner was subsequently attacked by law enforcement. She was later arrested for police interference and for opening her business during the COVID-19 shutdown. The video footage of the attack incited country-wide outrage.

Surviving the brutal encounter, the Black woman bar owner during a television interview with Fantástico—where she remained unidentified and her face unshown—stated that during the attack, "The more I struggled, the more he [the police officer] pushed on my neck" (France24 2020). Moreover, she made a direct African Diasporic symbolic and political connection to herself and George Floyd stating: *"I thought I was going to die like him"* (France24 2020) (My emphasis).

## 2. Methodology

Utilizing Black feminist historian and literary scholar Saidiya Hartman's theorization of the "afterlife of slavery", which is characterized by the sustained and generational "... skewed life chances, limited access to health and education, premature death, incarceration, and impoverishment" formulated and conditioned by slavery's "... racial calculus … and political arithmetic" (Hartman 2007, p. 6), this work interrogates the afterlife of Floyd and the afterlives of Black Brazilian victims and survivors of racial authoritarianism. Offering a brief yet comprehensive comparative analysis of historical and contemporary racial authoritarianism in Brazil and the U.S., this work examines how enslavement and the processes of racialization shape the contours of Black life, specifically through multiform anti-Black violence. With Brazil and the U.S. housing the largest Black populations in the Americas—and with Brazil housing the largest Black population in the African Diaspora—these two countries offer important observations on the enduring legacies of racialized violence and terror.

Several scholars have offered critical interventions on authoritarian power and racial authoritarian governance in Brazil and in the United States (Schwarcz 2022; González 2021; Weaver and Prowse 2020; Parker and Towler 2019; Blakeley 2012). According to Christopher C. Towler and Christopher Sebastian Parker, authoritarianism can be characterized by macro-and-micro analyses. The authors define authoritarianism from the macro perspective as "... in the context of comparative politics, in general, describes a regime type in which the power to govern is concentrated in a single party or run by a single figure. Authoritarian regimes are characterized by, among other things, weakened institutions, the unregulated use of executive power, repression, and patronage with its concomitant loyalty to the ruler or ruling party" (Parker and Towler 2019, p. 504). The micro analysis of authoritarianism "... consists of the culturally dominant group attempting to impose its beliefs on subordinate groups" (p. 504). Using slavery and the Jim Crow South as examples, the authors argue People of Color (POC) are not tolerated at macro-and-micro levels of authoritarianism due to their outsider status as non-White, and, therefore, deemed as non-American (p. 504).

Like Towler and Parker, Vesla M. Weaver and Gwen Prowse examine how racial authoritarianism has maintained the subordination of Black communities in the U.S. (Weaver and Prowse 2020, p. 1176). Defining racial authoritarianism as "… state oppression such that groups of residents live under extremely divergent experiences of government and laws" (p. 1176), the authors emphasize that police violence has been central in enforcing undemocratic practices against Black communities. Lilia Moritz Schwarcz discusses the intersections of authoritarianism against Black, Indigenous, LGBTQ+ and women communities in Brazil during the presidency of right-wing, racist, and sexist Brazilian populist, Jair Bolsonaro, and how his administration was symptomatic of the country's legacy of imperialism, enslavement, and military dictatorship (Schwarcz 2022, pp. 12–13). For this work, I utilize Weaver and Prose's definition of racial authoritarianism. I also utilize Towler and Parker's macro-and-micro analyses of racial authoritarianism to showcase how racial violence against Black communities in the U.S and Brazil is not only

perpetuated at the legislative level by state actors and agents of the state (i.e., the police) but is enacted by community members, who, through social conformity, impose the racialized order of the state; thus, reinforcing the subjugation of Black life (p. 511). Like Schwarcz, I also analyze racial authoritarianism from multiform frameworks, citing Kimberlé Crenshaw's theory of intersectionality (Crenshaw 1989). Here, I examine how racial authoritarianism impacts Black communities in Brazil and in the U.S., across race, gender, class, and sexuality.

Moreover, this work examines the murder of Floyd as a part of African Diasporic understandings of linked fate in the U.S. and in Brazil. Coined by Michael C. Dawson, linked fate or the Black utility heuristic supposes that because of racial and economic oppression in the U.S., Black Americans measure their fates or life outcomes as affected by or determined by the fate of the Black race as a whole (Dawson 1994). Although Dawson's advancement of linked fate specifically focuses on the political behavior of Black Americans, I utilize a qualitative analysis of linked fate to assess the political and symbolic linkages between Black Americans and Black Brazilians as victims and survivors of racial authoritarianism. Using case studies of Black American and Black Brazilian victims and survivors of racial authoritarianism as a way to showcase linked fate and the afterlife of slavery in the respective countries, this work addresses the aforementioned by examining the afterlife of Floyd in Brazil.

### 3. Constructing Racialized Bedrocks of Authoritarianism: The Racial Apparatuses of Anti-Black Violence in Brazil and the United States

Racial authoritarianism in Brazil and in the U.S. find origins in the legacies of enslavement and the de jure and de facto processes of racialization and racial formations that render Black people fungible, non-citizen, and, therefore, non-human (Omi and Winant 1986; Snorton 2018). Colonized by Portugal, Brazil was the last country in the Western hemisphere to abolish slavery by the *Lei Áurea* (the Golden Law) signed by Princess Isabel on 13 May 1888–23 years after the U.S. abolished slavery in 1865. While scholars have brought forth various estimates about how many Africans were captured and brought to Brazil, the exact numbers are unknown (Aidoo 2018, p. 13). However, as Lamonte Aidoo notes, "… it is estimated that from the beginning of the slave trade to the abolition of slavery in 1888, Brazil imported close to four million slaves—the largest number of any country in the Americas. The total number of slaves brought to British North America between 1701 and 1860 has been estimated at five hundred thousand" (Aidoo 2018, p. 13).

Slave patrols in Brazil and the U.S. were one of the primary systems in maintaining racial authoritarianism, as they "… functioned to capture fugitives, police the behavior and movement of enslaved Africans, prevent resistance efforts, and protect the institution of slavery" (Wilson 2022). With slave patrols in the U.S. being known as bounty hunters, paddy rollers, and night watchers and in Brazil, as pedestrian guards, according to Betty L. Wilson, "… by the end of the 18th century [in the U.S.], "slave patrols were in every slave state in the country" (Wilson 2022). In the state of Bahia, located in the northeastern part of Brazil, where Salvador, its capital city, was the former colonial capital of Brazil and a major slave port, "slave patrols were often charged to raid *quilombos* [maroon communities, my emphasis] for runaway slaves" (Wilson 2022). While the racial demographics of slave patrollers differed in the U.S. and Brazil—with slave patrollers in the U.S. predominantly being White men and in Brazil, slave patrollers consisting of White, Indigenous, and enslaved and freed Blacks and mulattos—slave patrols operated "to not only detain fugitive slaves but to also administer brutal punishment to runaways such as whippings, mutilation, and lynchings—and did so with virtual impunity" (Wilson 2022).**Error! Reference source not found.**

The legacies of slave patrols in Brazil can also be found in the country's formation of death squads or *milícias*—illegal paramilitary groups consisting of former and current civil and military police paid for hire—which according to Christen A. Smith, developed under

the military dictatorship from 1964 to 1985, where political organizing was repressed (Smith 2017, p. 125). Although Brazil underwent a period of re-democratization in the late 1980s, Yanilda María González argues the country's police force and its modes of repression overwhelmingly stayed the same (González 2021, p. 8). Moreover, González emphasizes that victims of police repression in Brazil and other Latin American countries are overdetermined by race and class and police forces in the region are "… the least likely to be held accountable by the judiciary" (p. 9). While Weaver and Prose would agree with González's assessment that Black and communities of color are disproportionately impacted by police violence and state agents are rarely held accountable at the judicial level (Ruge 2021), they advance that "unlike Latin American cases, where authoritarian practices predated and then survived democratic openings, in the United States, authoritarian policing tended to develop after democratic expansions" (Weaver and Prowse 2020, p. 1178).

Regarding the processes of racialization in Brazil, the ideological and political articulation of the myth of racial democracy undergirded racial formations in the country. Widely attributed to Brazilian sociologist Gilberto Freyre in his 1933 book, *Casa-Grande e Senzala* (The Masters and the Slaves), racial democracy promoted that because of *mestiçagem* (miscegenation) of Amerindian (Indigenous), African, and European peoples, race relations in Brazil were harmonious and less violent in comparison to the U.S., which had strict racial segregation laws such as Jim Crow. Instead of race and racism being factors that demarcated social and political status, racial democracy promoted class as the determining factor of inequality. Here, race/color caste systems vis-à-vis *mestiçagem*—"which asserts that race mixture has made racial identification a very indeterminate and unnecessary practice" (Hernández 2015, p. 685)—was emphasized, instead of "static" understandings of race in the U.S.[Error! Reference source not found.]

Despite the promotion of racial democracy and *mestiçagem* throughout the nineteenth and twentieth centuries, the Brazilian State would resent its majority Black population and employed *branqueamento* (Whitening), which according to Tanya Katerí Hernández, "…refers to both the aspiration and possibility of transforming one's social status by approaching Whiteness. An individual can become socially lighter by marrying a lighter-skinned partner or by becoming wealthy or famous" (Hernández 2015, p. 685). Mulatto or lighter-skin, mixed-raced Brazilians have historically been favored over darker-skinned Black Brazilians across socio-economic and political lines; however, "… have fewer privileges than the numerical minority of empowered Whites" (p. 686). After the abolition of slavery, the Brazilian Government utilized *branqueamento* by "… clos[ing] the country's borders to African immigrants, den[ying] [B]lack Brazilians the rights to lands inhabited by the descendants of runaway slaves and subsidized the voyage of millions of German and Italian workers, providing them with citizenship, land grants, and stipends when they arrived" (Aidoo 2018, p. 22).

Black Brazilian scholar-activists such as Sueli Carneiro, Abdias Nascimento, Lélia Gonzalez, and Beatriz Nascimento challenged the Brazilian State's contradictions and fallacies of racial democracy, noting that Black Brazilians, pre-and-post slave abolition, were marginalized across political, social, and cultural lines. This can also be applied to the U.S., as leaders such as Frederick Douglass, Fannie Lou Hamer, Dr. Martin Luther King, Jr., and Malcolm X challenged the U.S. government's continued denial and intolerance of Black people, despite the government professing that "all men are created equal." Here, I offer that Hartman's afterlife of slavery not only showcases the sustainment of socio-economic and political disparities and racial authoritarian violence in the U.S. and Brazil, as "… [B]lack lives are still imperiled and devalued…" (p. 17) but also highlights contestations of what constitutes Black citizenship, humanization, freedom, and inclusion in the respective countries (Hartman 1997, pp. 5–7).

### 4. "Here, We Have a George Floyd Every 23 Minutes": Modern-Day Racial Authoritarianism and the Precarity of Black Life in Brazil

When the video footage of George Floyd's brutal murder was posted on Facebook and Instagram by Darnella Frazier, a Black woman bystander, nationwide and global protests ensued in the U.S., Canada, Nigeria, Italy, France, United Kingdom, and countless other countries. Utilizing Floyd's dying words, "I can't breathe" and the #BlackLivesMatter hashtag as a rallying cry, the protests were in solidarity with calls for justice for Floyd in the U.S. by the Black Lives Matter Movement and the Movement for Black Lives. In Brazil, protests in solidarity with Floyd also occurred, and just like in other countries, murals and artistic pieces were created as a symbolic ode to his life and legacy. However, seven days prior to Floyd's death, Black Brazilians were having a racial reckoning of their own, in response to the shooting death of 14-year-old Black teenager, João Pedro Matos Pinto, in the São Gonçalo's Complexo do Salgueiro, a *favela* (slum or shanty town) in Rio de Janeiro, on 18 May 2020, (Phillips 2020; Watson 2020).

Unable to attend school due to the COVID-19 lockdown, João Pedro was in the house with his cousins when 10 officers burst into the home, allegedly looking for drug traffickers (Freelon 2020; Phillips 2020). The police fired more than 70 shots in the home. João Pedro was shot in the back, and later, airlifted into a police helicopter. However, after many hours, the police refused to give João Pedro's family any information about his whereabouts, prompting his cousin to send a message on Twitter using the hashtag #procurasejoaopedro (find João Pedro) in efforts to locate him (Freelon 2020). While the hashtag went viral on social media and João Pedro's disappearance made headlines, sadly, the teenager's body was found 17 h later in the public morgue.

In response to the lack of police accountability for João Pedro's death, "… on 25 May 2020, the same day George Floyd was killed, a coalition of Black movement groups led a massive digital protest –#JustiçaporJoãoPedro. The next day, lawyers working on behalf of Black and favela social groups filed a petition with Brazil's Supreme Court to halt operations for the rest of the COVID-19 pandemic" (Freelon 2020).[4] While the police said they were investigating his case, no one was arrested for João Pedro's murder. In fact, the police officers responsible for his death were still permitted to conduct police raids (Ruge 2021).

Unfortunately, João Pedro was not the only young, Black Brazilian male whose life was taken by police in Rio on May 18th. Twenty-one-year-old Iago César dos Reis Gonzaga was tortured and murdered by the police (Freelon 2020). That same week, two young, Black men—18-year-old João Vitor Gomes da Rocha and 19-year-old Rodrigo Cerqueira—were also killed by the police (Freelon 2020).

In April 2021, police officer Derek Chauvin was convicted of the murder of Floyd. While many were relieved by the outcome, Rafaela Matos, João Pedro's mother, in an interview with *American Quarterly*, made a Diasporic linked fate connection to Floyd and the lack of justice for her son: "I see the George Floyd case and I feel even more anguish. I ask myself, why was the case resolved and not this one?... João Pedro was not the first and he will not be the last, unfortunately. No mother should expect to lose her son like this, and that is why we are seeking justice, so that this impunity ends." In the same interview, Marcelle Decothé, the former program manager of the Marielle Franco Institute[5], also stated: "Police violence here gets normalized in such a way that the deaths of countless George Floyds here get no mention in local and national news" (Ruge 2021).

Here, I offer that Matos and Decothé's responses to Floyd's death, as it relates to João Pedro's murder and other instances of racial authoritarian violence in Brazil, reflect African Diasporic understandings of Dawson's linked fate for various reasons. Linked fate primarily focuses on Black American racial-group interests as it relates to racial discrimination. Positing that because racial discrimination disproportionately impacts Black people in the U.S., this subordination elicits a linked fate response—"a feeling as if one's own fate is affected by what happens to other category members and/or the group as a whole" (Monk 2020; Dawson 1994). Matos and Decothé's statements not only showcase a Black

political consciousness as it relates to João Pedro and Floyd as category victims of racial authoritarian violence but also how, due to the afterlife of slavery, this violence has and continues to subordinate Black groups in Brazil and in the U.S. as a whole. Their responses also offer insights on linked fate as it relates to police accountability and the importance of Black Brazilians and Black Americans uniting to organize collectively for justice for Black victims of racial authoritarian violence, due to shared histories of and struggles against racial discrimination. Although in the U.S. and in Brazil, police officers are rarely held accountable at the judicial level for racial authoritarian violence[6] (González 2021, p. 9), Matos perceives her son's death as politically and symbolically akin to Floyd's, thus, believing both deserve justice (Amparo 2020).

Moreover, given that Brazil houses the largest Black population in the African Diaspora and data show that police officers murder Black people at a rate three times more than in the U.S., Decothé' noted how the deaths of Black Brazilians do not get enough local and national media attention in the country. I also argue that while the death of Floyd was able to garner international media attention, outrage, and solidarity, the murders of Black Brazilians do not often elicit such global support. Here, Matos and Decothé's responses also bring to the fore Diasporic political attitudes on who is (and who is not seen) as a category member of a racialized group, and who is "worthy" of rallying for justice (Cohen 1999).

Like Matos and Decothé, countless Black Brazilians have made Diasporic linked fate connections to anti-Black police violence in the country to that of Floyd's death. Noting that what is happening in the country is a *genocído do povo negro* (genocide of Black people), in response to João Pedro and other acts of racial authoritarian violence, thousands took to the streets and social media using the slogan, "*Vidas Negras Importam*" (Black Lives Matter in Brazilian Portuguese) (Freelon 2020). Inspired by the Black Lives Matter Movement in the U.S., in 2017, United Nations Brazil spearheaded the *Vidas Negras* (Black Lives) campaign, which was launched during Brazil's Black Consciousness Month in November. The campaign emphasized the stark statistic that a young Black man dies every 23 min in the country (United Nations Office on Drugs and Crime n.d.). In comparison, it is estimated that police kill about 1200 people a year in the U.S., with Black people comprising about 27 percent of those police murders, despite constituting only 13 percent of the population (Mapping Violence in U.S. Report). Data show that in Brazil, police kill more than 6000 people per year. In 2022, Black people were 83 percent of those deaths, even though they constitute more than half—56 percent—of the population (Carvalho and Costa 2023).

The murders of young Black men from the *favelas* are so sustained that research "… conducted by Special Secretariat for Policies for the Promotion of Racial Equality (SEPPIR) and by the Federal State [showed] that 56 percent of the Brazilian population agrees with the statement that  the violent death of a [B]lack young person shocks society less than the death of a [W]hite young person" (United Nations Office on Drugs and Crime n.d.). Here, Weaver and Prose's definition of racial authoritarian violence applies to the Brazilian context, as Black communities—in this case, young, poor, Black males—are dying under disparate circumstances in comparison to White Brazilians (p. 1176). Similar to Black Americans, class also plays an important factor in racial authoritarian violence, as the majority of the murders of Black males in Brazil occur in the *favelas* (González 2021, p. 9; Dawson 1994, p. 13)**.** However, as Hartman advances, the declaration of the abolition of slavery did not dismantle the racial logic and mechanisms of the institution, and, therefore, the objectification and dehumanization of Black people persist under so-called democratic values. Based on the "skewed life chances … and premature death" (Hartman 2007, p. 6), of Black Brazilians, the lyric from Black Brazilian singer, Elza Soares, from her 2002 anti-racist song, "*A Carne*" (The Meat) applies to the historical and present-day plight of Black Brazilians: "The cheapest meat on the market is Black meat" (Davison 2022).

While the aforementioned examples are what Towler and Parker describe as macro-analyses of racial authoritarianism, it is also imperative to examine how micro-analyses of racial authoritarian violence are enforced by community members who reify the anti-

Black racialized hierarchy of the State. On 24 January 2022, Moïse Mugenyi Kabagambe, a 24-year-old Congolese migrant, went to a seaside bar located in the Barra da Tijuca neighborhood, near one of Rio's most popular beaches, "… to demand 200 reais⁷ in unpaid wages … where he had worked informally as a waiter serving beach-goers" (Barretto Briso and Phillips 2022). Instead of payment, Kabagambe was brutally beaten by three White men, who were his colleagues. No bystanders intervened, nor did anyone call the police. Kabagambe died from his injuries (McCoy and Pessoa 2022). Moreover, Kagabame's death occurred a few minutes away from a home owned by Brazil's former, right-wing conservative president, Jair Bolsonaro, who, during that time, was in the last year of his presidential term. Nicknamed "Trump of the Tropics", Bolsonaro said nothing about his murder (Barretto Briso and Phillips 2022).

In 2011, Kabagambe, along with his mother and siblings, fled to Brazil from the city of Bunia in the Democratic Republic of the Congo after his grandmother was killed and his father disappeared due to ethnic-group conflict in the country (Jeantet and Rodrigues 2022). Here, as Towler and Parker suggest, Kabagambe's status as a dark-skinned, poor, African migrant was in opposition to Brazilian norms and values that center Whiteness and, therefore, his request for payment was deemed intolerable and punishable by death (p. 504).

In response to Kabagambe's murder, Douglas Belchior, an activist with Coalizão Negra Por Direitos, made a linked fate connection to not only Floyd but to the afterlives of slavery that continue to subordinate Black life in Brazil:

> "The barbaric murder of [B]lack people is shamefully commonplace in Brazil … Here, we have a George Floyd every 23 min. We have a Moïse every 23 min. We are constantly being killed … Brazil is a country built on slavery. It's the country that endured slavery for the longest, was the last in the Americas to abolish it—and the way Brazilian society was organized post-slavery was designed to perpetuate the social dynamic by which [B]lack people were subjected to the slave owner's way of thinking" (Barretto Briso and Phillips 2022).

## 5. *Whose Black Life Matters?* George Floyd and Intersectional Racial Authoritarianism in Brazil and the U.S.

However, macro-and-micro analyses on racial authoritarianism have often centered Black cis-heterosexual men as the only victims of anti-Black violence (Crenshaw 1989) (Ritchie 2017). While Black men in Brazil and the U.S. are disproportionately impacted by racial authoritarian violence, Black women, queer, transgender, and gender-expansive communities are overwhelmingly victims of violence, too. As mentioned previously, linked fate offers that Black Americans view themselves as "linked" or bound by quotidian experiences of racial discrimination. However, as Cathy J. Cohen critiques, when the consensus issue of race intersects with cross-cutting issues such as gender and sexuality, understandings of linked fate weaken as a shared Black political agenda (Cohen 1999, p. 9).

As I address in the previous section with Matos and Decothé's statements on justice for João Pedro and Floyd, Cohen's critique calls into question who is seen as a category member of a racialized group and who is deemed an outsider. In her work, the *Boundaries of Blackness: AIDS and the Breakdown of Black Politics* (Cohen 1999), Cohen offers that certain groups within the Black community were denied access to Black political agendas due to sexism, classism, respectability politics, and queerphobia. Here, Cohen's critique of linked fate and Kimberlé Crenshaw's theory of intersectionality, which examines how race, class, gender, and sexuality overlap, bring to the fore how Black women, and queer and transgender communities in the U.S. and in Brazil are overlooked as victims and survivors of racial authoritarian violence, especially in the context of justice for Floyd and his afterlife.

For example, on 13 March 2020, Breonna Taylor, a 26-year-old Black American woman, was fatally shot in her apartment in Louisville, Kentucky. Taylor, an emergency medical technician and first responder, was sleeping in her apartment with her boyfriend, Kenneth Walker, when Louisville Metro Police executed a "no knock warrant", "looking for drugs they never found, reportedly trafficked by a person who did not live with Breonna or in her complex–and whom they already had in custody" (Justice for Breonna Taylor n.d.). Police officers shot in her home over 20 times, shooting Taylor eight times. Although Taylor's murder occurred two months prior to Floyd's, calls for justice for Taylor did not gain the same attention. During protests for Floyd, many Black women activists questioned why Taylor's murder was disconnected from the mainstream rallying cry for justice (Gupta 2020). As author and activist, Andrea Ritchie noted, "We're not trying to compete with Floyd's story, we're trying to complete the story" (Gupta 2020). However, it was initiatives such as the #SayHerName campaign, spearheaded by Crenshaw, which "brings awareness to often invisible names and stories of Black women and girls who have been victimized by racist police violence", that brought Taylor's story to mainstream media (African American Policy Forum n.d.).

In Brazil, Black women are disproportionately impacted by racial authoritarian violence. Months after the murder of Floyd, Black Brazilian feminist, activist, and human rights defender, Jane Beatriz Machado da Silva, 60, was murdered after police invaded her home in Grande Cruzeiro, a poor neighborhood in the city of Porto Alegre, in the state of Rio Grande do Sul (Human Rights Defender Memorial n.d.). On 8 December 2020, police raided da Silva's home without a court warrant. Protesting the illegality of their search, during her encounter with police, da Silva died. Although police claim she died by natural causes due to a brain hemorrhage, her neighbor "… claim[ed] to have witnessed the incident, stat[ing] that the police head beaten Jane Beatriz, which brought on the hemorrhage" (Human Rights Defender Memorial n.d.). While over 300 civil society organizations in Brazil signed and issued a press release demanding an investigation for her death, da Silva's murder, in comparison to Floyd and João Pedro's, is relatively unknown (Conectas 2020).

Black trans people in Brazil and the U.S. are victimized at the intersections of anti-Black and anti-transgender authoritarian violence. Brazil has the highest rates of transgender murders in the world—with 41 percent of all anti-transgender murders occurring in the country (Benevides and Nogueira Bonfim Naider 2019, p. 8). The majority of victims—82 percent—are Black (Benevides and Nogueira Bonfim Naider 2019, pp. 17, 21; Borges 2019; Swift 2018). Although Black Brazilian feminist-activist Erika Hilton became the first transgender councilor elected by the city of São Paulo in November 2020, that same year, 175 transgender women were murdered in Brazil, which was a 41 percent increase from the year prior (Sudré 2021; May and Guima 2021).

According to the Human Rights Campaign (HRC), in the U.S., there were a total of 44 fatalities of transgender people in 2020. In 2021, the number of deaths was 59, one of the highest murders of transgender deaths on record in the country (Human Rights Campaign n.d.). The majority of victims were Black and Latinx transgender women; however, Black trans men are also targets of racial authoritarian violence. Two days after the murder of Floyd, Tony McDade, a 38-year-old Black American transgender man, was shot and killed by a police officer in Tallahassee, Florida on 27 May 2020. According to Tallahassee police, McDade was a suspect in the fatal stabbing of 21-year-old Malik Johnson (Mahoney 2021). Upon arriving at the scene, police stated they found a bloody knife and that McDade pointed a firearm at police. However, a neighbor of McDade's, who witnessed the encounter, said, "I never heard, Get down, freeze, I'm an officer.' I never heard nothing. I just heard gunshots" (Stanford Libraries n.d.).

A week after Floyd's death and less than a few days after McDade's murder, Iyanna Dior, a 21-year-old Black transgender woman was attacked by a mob at a Minneapolis gas station after a car accident on 1 June 2020. While protests for justice for Floyd were occurring in Minneapolis and nationwide, a video was shared on social media showing Dior

and "… an unidentified Black man having a disagreement outside of a convenience store before a large group begins beating Dior and yelling transphobic slurs" (Walsh 2020). The video shows Dior being beaten by 15–30 people, reportedly all of the assailants being Black cisgender men (Walsh 2020). A few days later, two Black transgender women—Dominique "Rem"Mie" Fells and Riah Milton were murdered. On 8 June 2020, Fells, 27, was brutally dismembered and murdered by an acquaintance, Akhenaton Jones, in Philadelphia, Pennsylvania. Her body was found floating in Schuykill River. Fells "had been stabbed repeatedly, with trauma to her face and head and both legs severed mid-thigh" (Avery 2020). In November 2020, Jones was arrested for her murder. On 9 June 2020, Milton, 25, was shot and murdered during a robbery in Liberty Township, Ohio (Carlisle 2020). Here, it is also important to note that the beatings and murders of Dior, Fells, and Milton all occurred during LGBTQ Pride Month in the U.S.[8]

In response to Dior's violent attack, Black transgender woman activist, Raquel Willis, tweeted:

"A young Black trans woman named #IyannaDior was brutally attacked by a group of mostly Black cis men in Minneapolis last night… But this is why we must always clarify that all #BlackLivesMatter every time we invoke this chant. What will happen to us Black women, Black LGBTQ+ folks, Black trans women if we continue to only rally around the state violence that cishet Black men face? The police, the state, and white supremacy are killing us, but so is an insecure version of masculinity that views Black womanhood, transness, and queerness as a threat. If you aren't rallying behind Black trans victims of violence as much as you are cishet Black men, you don't really believe that Black Lives Matter" (Sanders 2020).

In a 2020 interview I conducted with Marry Ferreira, a Black Brazilian journalist and co-founder of the Kilomba Collective, the first Black Brazilian immigrant women's collective in the U.S., she also shared Willis' sentiments on the radical imperative of an intersectional, linked fate approach to challenging racial authoritarianism. Moreover, Ferreira also emphasized that solidarity and collective struggles to end anti-Black racial violence must extend to all Black people—and not just Black communities in the U.S.:

"The Brazilian crisis of police violence cannot be separated from the context of anti-Blackness and the genocide of Black people that is happening in Brazil. We can't really disconnect this fight against State violence from one context to another. The context may be different, and the strategies of the State may be different, but our people are still dying. I really don't think that one of us can be free if none of us are free. That includes all of us in all these spaces and it includes all of us in other countries… So, when we hear about João Pedro, Miguel Otávio, Nina Pop, and Breonna Taylor, we can't forget that."

"When I see the African-American community thriving, I thrive too. And when I feel like we lost someone, I feel that too because this is what international solidarity means. We can't just continue to keep going if our people are just dying in other places… We got justice for George Floyd because the whole world was saying justice for George Floyd. But what about Claudia da Silva Ferreira? What about so many other Black women that we don't know?" (Swift 2020).

Here, Willis and Ferreira's statements on how Black transgender, queer, and women communities are overlooked as victims of racial authoritarian violence by the government, police, and community members echoes Cohen's critique of linked fate and how certain groups in the Black community endure a secondary marginalization due to their outsider status as non-male, non-normative, non-citizen, non-American, etc., (Cohen 1999, p. 9). Moreover, recent research conducted by Tehama Lopez Bunyasi and Candis Watts Smith on the Black Lives Matter Movement, the Movement for Black Lives, and Black people's perceptions of marginalized, Black communities in the U.S., upholds Cohen's assertion on secondary marginalization and linked fate. Analyzing data from the 2016 Collaborative

Multi-Racial Post-Election Survey, which include responses from more than 3000 Black people, Lopez Bunyasi and Smith's research shows that although organizations such as Black Lives Matter and the Movement for Black Lives have an intersectional approach to justice for Black people, respectability politics "… leads Black people to disengage from supporting Blacks who have been marginalized" (Lopez Bunyasi and Smith 2019).

## 6. Conclusions: The Aftermath of George Floyd in the U.S. and in Brazil

While protests and resistance efforts for justice for Floyd brought more exposure to address racial authoritarianism in the U.S., and in Brazil, overwhelmingly, the exposure and change were temporary and symbolic. In 2021, the George Floyd Justice in Policing Act (H.R.1280) was proposed and sponsored by the House Committee on the Judiciary in the U.S. The bill "addresses a wide range of policies and issues regarding policing practices and law enforcement accountability. It increases accountability for law enforcement misconduct, restricts the use of certain policing practices, enhances transparency and data collection, and establishes best practices and training requirements" (U.S. Congress 2021). Moreover, the bill banned the use of chokeholds and no-knock warrants, which claimed the lives of Floyd and Taylor (U.S. House Committee on the Judiciary n.d.). While the bill was passed in the House, it did not pass in the Senate.

In May 2023, the Brazilian and U.S. governments re-established their bilateral agreement, the U.S-Brazil Joint Action Plan to Eliminate Racial and Ethnic Discrimination and Promote Equality (JAPER) (Johnson and Batista 2023). Formulated in March 2008, JAPER "was the first ever bilateral agreement to specifically target racism" in the respective countries (Johnson and Batista 2023). In congruence with ideals on linked fate, it was the deaths of Floyd and Black Brazilians that ignited Black Brazilian organizations to demand the revival of the agreement. On 8 February 2023, 10 Black Brazilian civil society organizations delivered a letter to Brazil's President Luiz Inácio Lula da Silva and U.S. President Joe Biden, two days before the pair were to meet in Washington, D.C. (Washington Brazil Office 2023). The letter outlined a list of demands, including that both countries need to prioritize JAPER to address disparities in education, gender, economic development, health, political representation, and violence against Black people (Washington Brazil Office 2023).

In December 2023, experts from the UN International Independent Expert Mechanism to Advance Racial Justice in the Context of Law Enforcement (which was created in 2021 in response to Floyd's death), visited the following Brazilian cities—the capital of Brazil, Brasilia, Salvador, Fortaleza, São Paulo, and Rio de Janeiro—to address the systemic murders of Black Brazilians by police. Meeting with civil society organizations, victims, and families, the delegation concluded that "Brazil's Government must end the brutal violence being inflicted on people of African descent by the country's police forces and hold perpetrators accountable for their crimes while ensuring justice for victims" (OHCHR 2023). Moreover, civil society and grassroots organizations in the U.S. and Brazil such as Black Women Radicals, the Marielle Franco Institute, the Kilomba Collective, the Marsha P. Johnson Institute, the Movement for Black Lives, and Odara-Instituto da Mulher Negra continue to organize and build solidarity against intersectional, racial authoritarian violence across the Americas.

However, despite these valiant efforts, the afterlives of slavery, George Floyd, and Black Brazilian victims and survivors of racial authoritarian violence still persist, with what seems to be no end in sight.

**Funding:** This research received no external funding.

**Institutional Review Board Statement:** Not applicable.

**Informed Consent Statement:** Informed consent was obtained from all subjects involved in the study.

**Data Availability Statement:** The original contributions presented in the study are included in the article, further inquiries can be directed to the corresponding author.

**Conflicts of Interest:** The authors declare no conflict of interest.

## Notes

1. In April 2021, Chauvin was found guilty of second-degree murder, third-degree murder, and second-degree manslaughter. In June 2021, he was sentenced to 22 ½ years in federal prison (Senter and Dewan 2022).
2. Given that slave patrols in Brazil also consisted of enslaved and freed Blacks and mulattos, this fact also reflects the contemporary demographics of the police force in the country. For example, in Bahia—the state that houses the largest Black population in Brazil—the majority of the police force is Black (Smith 2017, p. 49).
3. For example, while the Brazilian Institute of Geography and Statistics (IGBE) lists the following racial-color categories on the Brazilian census—*branco* (white), *pardo* (Brown or a mixed-raced person with African ancestry), *preto* (Black), *amarelo* (yellow), and *indio* (Indigenous)—there are over 136 racial-color categories that are used in the country based on physical traits, hair type, skin color, socio-economic and/or educational status, and more (Monk 2016).
4. In June 2020, Brazil's Supreme Court prohibited police raids in Rio de Janeiro's favelas during the COVID-19 pandemic, "except in absolutely exceptional cases" (Reuters 2020).
5. The Marielle Franco Institute is an organization with a mission to receive justice for Marielle Franco, a Black Brazilian bisexual-feminist and councilwoman from Rio assassinated by former police officers in 2018.
6. For example, as Thiago Amparo highlights, "… [i]n 20 years of work with the Minneapolis Police Department, Derek Chauvin—the police officer who asphyxiated George Floyd with his knees—had already faced at least 17 misconduct complaints, with virtually no consequences for him" (Amparo 2020).
7. 200 reais is approximately $39.07 USD.
8. In the United States, LGBTQ Pride Month is celebrated each year in the month of June, in honor of the 1969 Stonewall Riots—also known as the Stonewall Rebellions—in New York City. On 28 June 1969, New York City police raided the Stonewall Inn, a gay club in Greenwich Village. Tired of police harassment, gay, lesbian, and transgender communities protested the police, over the course of six days. The Stonewall Riots are considered to be the start of the LGBTQ movement in the U.S. (Steinmetz 2019).

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
