# Peer review of "“I Thought I Was Going to Die like Him”: Racial Authoritarianism and the Afterlife of George Floyd in the United States and Brazil"

_socsci, doi:10.3390/socsci13060299_

Round 1
Reviewer 1 Report
Comments and Suggestions for Authors
See complete review report on overly wordiness of sentences.
Reviewer 2 Report
Comments and Suggestions for Authors
Dear Author,
Please see my comments as written to the Editor.
Thank you for the opportunity to review this manuscript. I think it addresses a really interesting and timely question: Do Black people subject to racialized authoritarianism see a collective struggle across country boundaries? While I appreciate the context that the author brings to bare to give us reason to compare the United States with Brazil, I do think the manuscript needs some work before it is published. With that being said, I think the authors should be allowed to revise and resubmit the manuscript. In what follows, I detail my thoughts and some questions about the manuscript as it stands:
My thoughts and questions include the following:
· I think the manuscript would benefit from some a bit of re-organization. From the introduction, it wasn’t clear to me what the author examines. This discussion comes up in the methodology suggestion.
· Speaking to my previous point about re-organization, it would be great to hear more about the framework being used to examine the data as well as what texts are brought to bare in the analysis. The author speaks to using Hartman’s theory here, but for those not familiar with Hartman’s work, it would be helpful to get a bit more detail about the theory and how it is being applied here. Moreover, the author engages in textual analysis – but doesn’t tell us about the text or the context of the case studies. I think more detail about what texts are brought to bare here and how the authors selected these particular cases for analysis would be helpful. In addition, the author brings in an interview that certainly helps to make author’s analytical case. However, as a reader, I wasn’t under the assumption that the author would bring in interviews here. Moreover, the author doesn’t bring in much analysis about the interview. I’m not sure if the author is under page constraints, but it would be worth it to delve deeper into the meaning of the interview and perhaps other interviews if the author conducted them.
· The author makes a strong case for why we should care the US to Brazil, which is important and helpful for the reader. However, the author doesn’t draw the throughline for why we might Black Americans and Black Brazilians to view themselves as part of a collective struggle. As a reader, I expected the author would draw out examples – not just of how police violence marginalizes Black people across borders – but of collective organizing or perceived collective struggle. This type of example doesn’t manifest until the discussion of the interview, but the interview isn’t fully discussed or analyzed. Since linked fate, as a theory, is associated with a behavior or feeling, it might be a good idea for the author to look to what motivated Dawson to develop the theory – shared experiences of discrimination at the hands of the state. I think the author has an opportunity to speak to a path dependency story (similar histories of slavery, similar experiences with state violence today) and opportunities for seeing a collective struggle based on these paths. At the moment, the connection to Dawson’s theory seems tenuous and not fully fleshed out.
o As an additional note, the author really doesn’t give us a discussion of how Hartman’s theoretical framework and Dawson’s theory work together in the author’s theorizing. I would like to see some stronger linkages made here.
Again, I think this is an interesting piece that is worthy of further development. I wish the author the best of luck in getting this work published.
Round 2
Reviewer 2 Report
Comments and Suggestions for Authors
I appreciate the effort that the author has put forth to speak to my comments. I think the article is greatly improved in its current form and should be published.